# Uptake of COVID-19 vaccinations amongst 3,433,483 children and young people: meta-analysis of UK prospective cohorts

Sarah J. Aldridge [1] ✉, Utkarsh Agrawal[2], Siobhán Murphy[3], Tristan Millington[4], Ashley Akbari [1], Fatima Almaghrabi[4], Sneha N. Anand[2], Stuart Bedston[1], Rosalind Goudie[2], Rowena Griffiths[1], Mark Joy[2], Emily Lowthian[5], Simon de Lusignan [2], Lynsey Patterson[3,6], Chris Robertson [7], Igor Rudan[8], Declan T. Bradley [3,6], Ronan A. Lyons [1], Aziz Sheikh[2,9] & Rhiannon K. Owen [1,9] ✉

SARS-CoV-2 infection in children and young people (CYP) can lead to life-threatening COVID-19, transmission within households and schools, and the development of long COVID. Using linked health and administrative data, we investigated vaccine uptake among 3,433,483 CYP aged 5–17 years across all UK nations between 4th August 2021 and 31st May 2022. We constructed national cohorts and undertook multi-state modelling and meta-analysis to identify associations between demographic variables and vaccine uptake. We found that uptake of the first COVID-19 vaccine among CYP was low across all four nations compared to other age groups and diminished with subsequent doses. Age and vaccination status of adults living in the same household were identified as important risk factors associated with vaccine uptake in CYP. For example, 5–11 year-olds were less likely to receive their first vaccine compared to 16–17 year-olds (adjusted Hazard Ratio [aHR]: 0.10 (95%CI: 0.06–0.19)), and CYP in unvaccinated households were less likely to receive their first vaccine compared to CYP in partially vaccinated households (aHR: 0.19, 95%CI 0.13–0.29).

Children and young people (CYP) spend a large proportion of their time in close proximity to one another and with challenging conditions for maintaining hygiene (e.g., shared school equipment or toys). These environments provide opportunities for the transmission for many pathogens – including SARS-CoV-2, the virus responsible for the COVID-19 pandemic. While CYP are less likely to suffer the severe consequences of COVID-19[1,2], there is still a risk of life-threatening

reactions such as Paediatric Multisystem Inflammatory Syndrome (PIMS)[3]. Additionally, CYP can transmit infection to others, particularly those more susceptible to the disease, such as older people or clinically vulnerable people. Behavioural methods such as social distancing, wearing masks, and hand washing have been shown to help reduce transmission. However, these methods are limited in slowing transmission on their own and are not always practical for these age

[1]Population Data Science, Swansea University Medical School, Faculty of Medicine, Health, and Life Science, Swansea University, Swansea, UK. [2]Nuffield Department of Primary Care Health Sciences, University of Oxford, Oxford, UK. [3]Centre for Public Health, School of Medicine, Dentistry and Biomedical Sciences, Queen's University, Belfast, UK. [4]Usher Institute, University of Edinburgh, Edinburgh, UK. [5]Department of Education and Childhood Studies, School of Social Sciences, Swansea University, Swansea, UK. [6]Public Health Agency, Belfast, UK. [7]Department of Mathematics and Statistics, Strathclyde University, Glasgow, UK and Public Health Scotland, Glasgow, UK. [8]Centre for Global Health, Usher Institute, the University of Edinburgh, Edinburgh, UK. [9]These authors jointly supervised this work: Aziz Sheikh, Rhiannon K. Owen. ✉e-mail: s.j.aldridge@swansea.ac.uk; r.k.owen@swansea.ac.uk

groups. As a result, most UK schools switched to remote learning formats to enable social distancing of pupils[4]. The policy decision to return CYP to school and in-person teaching following lockdown was of key importance for their own educational and social development, mental health and the benefit of their guardians and families[5–7]. For a combination of these reasons, pharmaceutical solutions have been implemented in the form of vaccinations for CYP in several age groups in the UK[8–10]. The Joint Committee on Vaccination and Immunisation (JCVI) is an independent expert advisory committee providing recommendations to the UK governments on vaccine effectiveness, safety, and scheduling. Following the initial success of the COVID-19 vaccine programme in adults and evidence from trials and observational studies on the benefits and risks of vaccines in CYP[11], the target population broadened to include younger adults with underlying health conditions and those who were living with vulnerable people, before expanding further to include all CYP. On 4th August 2021, JCVI recommended including 16–17 year-olds who were not in a clinical risk group in the Pfizer-BNT162b2 vaccine schedule[12], followed by similar recommendations for 12–15 year-olds on 13th September 2021[13] and 5–11 year-olds on 16th February 2022[9]. Each nation took time to review this advice before officially moving forward with the vaccination programme of each age group, making vaccinations available for these groups in General Practice surgeries and vaccination centres. In all cases the schedule started within 0–2 days following JCVI recommendation for 16–17 year-olds, 1–34 days for 12–15 year-olds and 14–47 days for 5–11 year-olds (see Supplementary Fig. 1 for nation-specific details).

The gradual rollout of the COVID-19 vaccine to progressively younger age groups of adults coincided with a steady decrease in uptake. Originally targeting older people and those with underlying health conditions, uptake was as high as 96% (80 years old and above receiving their primary schedule) in England, but declined to 70% in 18–19 year-olds[14]. Vaccination in CYP is complex and tightly bound to issues regarding consent, autonomy and access[14]. Within the UK COVID-19 vaccine uptake in CYP has been shown to vary in state-school pupils with region and ethnicity with a slower uptake in pupils who spoke english as a second language, attended special needs schools, received free school meals or lived in more deprived areas[15]. Outside of the UK, studies have shown that uptake of the COVID-19 vaccine in CYP can be influenced by several factors, including availability[16], region[17,18], age[17–19], ethnicity[17,18,20], immigration status[19], urbanicity[21], vulnerability[22], education[18], household income[18,19], parental vaccination status[18], prior COVID-19 diagnosis[18] and parents age[18].

In the US, it has been shown that during the time period of 1st August 2021 to 31st July 2022, COVID-19 was a leading cause of death in CYP. Crude death rates were estimated as 0.4 per 100,000 CYP aged 5–9 years, 0.5 per 100,000 CYP aged 10–14 years, and 1.8 per 100,000 for CYP aged 15–19 years[23]. It is known that overall mortality and morbidity differed with alternative dominant variants[24]. In the UK, the dominant variant from 1st August 2021 to 19th December 2021 was the Delta variant and from 20th December 2021 onwards was the Omicron variant[25].

Analysis of COVID-19 vaccine uptake is key to identifying influential factors and how these may be affecting different demographics. We aimed to investigate the uptake of the COVID-19 vaccines across all four UK nations following JCVI recommendation for CYP using prospective population-based cohort analyses on routinely collected electronic health record (EHR) data adjusting for infection as a competing risk, and to explore the association of vaccine uptake with age, sex, and household factors.

## Results

In England, 34% (744,763 of 2,180,740) of CYP received their first vaccine, 20% (442,631) received their second, and 2% (45,382) received their booster, leaving 66% (1,435,977) of CYP unvaccinated between 4th August 2021 and 31st May 2022 (Table 1 and Fig. 1). For the same period, Northern Ireland had the lowest uptake of the four nations where 24% (76,144 of 318,437) of CYP received their first vaccine, 11% (36,141) received their second and 1% (2862) received their booster, leaving 76% (242,293) unvaccinated. Scotland had the highest vaccine uptake where 46% (263,912 of 569,971) of CYP received their first vaccine, 25% (142,476) received their second and 2% (12745) received their booster, leaving 54% (306059) unvaccinated. In Wales, 37% (136,098 of 364,335) of CYP received their first vaccine and 25% (92,743) received their second dose. Wales had the highest uptake of the booster vaccine where 5% (19,387) of CYP received their booster, leaving 63% (228,237) unvaccinated. A summary of the cohort characteristics by each nation are provided in Supplementary Table 1.

In total 3,433,483 CYP were included in the study. Across all four nations, 35% (1,220,917) received their first vaccine, 21% (713,991) received their second, and 2% (80,376) received their third or booster vaccine between 4th August 2021 and 31st May 2022 (Table 1 and Fig. 1). During the study period, 13% (451,617) tested positive for infection with COVID-19, and <0.0001% (133) died. There were substantial differences in vaccine uptake by different age groups. The highest uptake was in the 16–17 year-olds where 71% (356,222) received their first vaccine, 54% (269,455) their second, and 14% (70,968) their booster. Uptake was lower in 12–15 year-olds of whom 59% (671,701) received their first vaccine, 39% (440,384) received their second and 1% (9408) received their booster. 5–11 year-olds had the lowest vaccine uptake, where 11% (192,994) received their first vaccine, and 0.2% (4152) received their second vaccine. 5–11 year-olds did not become eligible for their booster dose before the conclusion of the study window. CYP residing in households with vaccinated adults had the highest uptake, with 42% (1,095,438) receiving their first vaccine during the study. Comparatively, only 23% (112,472) and 4% (13,007) of CYP living in a partially vaccinated households and unvaccinated households received a vaccine, respectively. Sex and household size showed much smaller variations in uptake. See supplementary material Table 1 for each national population breakdown.

### Results by nation

In England, males were less likely to receive their first (aHR 0.97, 95%CI 0.96–0.97), second (aHR 0.98, 95%CI 0.98-0.99) and booster vaccines (aHR 0.97, 95%CI 0.95–0.98) compared to females (Table 2, Fig. 2). Uptake of first dose was less likely in 12–15 year-olds (aHR 0.72, 95%CI 0.72–0.72) and 5–11 year-olds (aHR 0.11, 95%CI 0.11–0.11) compared to 16–17 year-olds (Table 2, Fig. 2). A similar association was found for second and booster doses. CYP in households of two (aHR 0.86, 95%CI 0.85–0.87) and households of five or more (aHR 0.89, 95%CI 0.89–0.9) were less likely to receive their first vaccine compared to households of three, whilst households of four were more likely (aHR 1.11, 95%CI 1.11–1.12) to receive their first vaccine. There appeared to be a similar trend for uptake of the second and booster vaccinations, however, there appeared to be no difference between households of four compared to households of three for uptake of the booster dose (aHR 0.99, 95%CI 0.97–1.01). CYP in households with unvaccinated adults were less likely to receive the first vaccine (aHR 0.11, 95%CI 0.11–0.11) compared to partially vaccinated households, whilst CYP living in households with fully vaccinated adults were more likely to receive the first vaccine (2.83, 95%CI 2.80–2.85). A similar association was shown for both second and booster doses (Table 2, Fig. 2). Cumulative incidence plots for CYP in England showed a slower uptake proportionally compared to the other nations, however this uptake was still increasing at the end of the study period (Fig. 3, Supplementary Fig. 2).

In Northern Ireland, males were less likely to receive their first (aHR 0.87, 95%CI 0.86, 0.88) and second (aHR 0.9, 95%CI 0.88-0.92) vaccination compared to females (Table 2, Fig. 2). However, there was insufficient evidence to detect a difference for booster dose (aHR 0.95, 95%CI 0.88–1.02). Similarly to England, 12–15 year-olds were less likely

**Table 1 | Cohort summary for each variable for all 4 UK nations : The counts for each variable for the total UK cohort, their proportion of the total population, and the proportion of each subgroup to receive each COVID-19 vaccine**

| | | n | % | No vaccine | % | 1st dose | % | 2nd dose | % | Booster dose | % |
|---|---|---|---|---|---|---|---|---|---|---|---|
| Total | | 3,433,483 | | 2,212,566 | 64.4 | 1,220,917 | 35.6 | 713,991 | 20.8 | 80,376 | 2.3 |
| Sex | Female | 1,675,102 | 48.8 | 1,072,047 | 64.0 | 603,055 | 36.0 | 355,531 | 21.2 | 40,721 | 2.4 |
| | Male | 1,758,381 | 51.2 | 1,140,519 | 64.9 | 617,862 | 35.1 | 358,460 | 20.4 | 39,655 | 2.3 |
| Age group | 05-11 years old | 1,802,237 | 52.5 | 1,609,243 | 89.3 | 192,994 | 10.7 | 4152 | 0.2 | 0 | - |
| | 12-15 years old | 1,132,106 | 33.0 | 460,405 | 40.7 | 671,701 | 59.3 | 440,384 | 38.9 | 9,408 | 0.8 |
| | 16-17 years old | 499,140 | 14.5 | 142,918 | 28.6 | 356,222 | 71.4 | 269,455 | 54.0 | 70,968 | 14.2 |
| Number of people in household | 2 | 241,069 | 7.0 | 155,360 | 64.4 | 85,709 | 35.6 | 50,899 | 21.1 | 6216 | 2.6 |
| | 3 | 701,384 | 20.4 | 442,598 | 63.1 | 258,786 | 36.9 | 155,709 | 22.2 | 19,455 | 2.8 |
| | 4 | 1,198,834 | 34.9 | 730,465 | 60.9 | 468,369 | 39.1 | 284,043 | 23.7 | 33,239 | 2.8 |
| | 5+ | 1,292,196 | 37.6 | 884,143 | 68.4 | 408,053 | 31.6 | 223,340 | 17.3 | 21,466 | 1.7 |
| Household vaccination status | Unvaccinated | 350,845 | 10.2 | 337,838 | 96.3 | 13,007 | 3.7 | 4490 | 1.3 | 288 | 0.1 |
| | Partially Vaccinated | 490,508 | 14.3 | 378,036 | 77.1 | 112,472 | 22.9 | 54,579 | 11.1 | 5,080 | 1.0 |
| | Fully Vaccinated | 2,592,130 | 75.5 | 1,496,692 | 57.7 | 1,095,438 | 42.3 | 654,922 | 25.3 | 75,008 | 2.9 |
| Nation | England | 2,180,740 | 63.5 | 1,435,977 | 65.8 | 744,763 | 34.2 | 442,631 | 20.3 | 45,382 | 2.1 |
| | Northern Ireland | 318,437 | 9.3 | 242,293 | 76.1 | 76,144 | 23.9 | 36,141 | 11.3 | 2862 | 0.9 |
| | Scotland | 569,971 | 16.6 | 306,059 | 53.7 | 263,912 | 46.3 | 142,476 | 25.0 | 12,745 | 2.2 |
| | Wales | 364,335 | 10.6 | 228,237 | 62.6 | 136,098 | 37.4 | 92,743 | 25.5 | 19,387 | 5.3 |

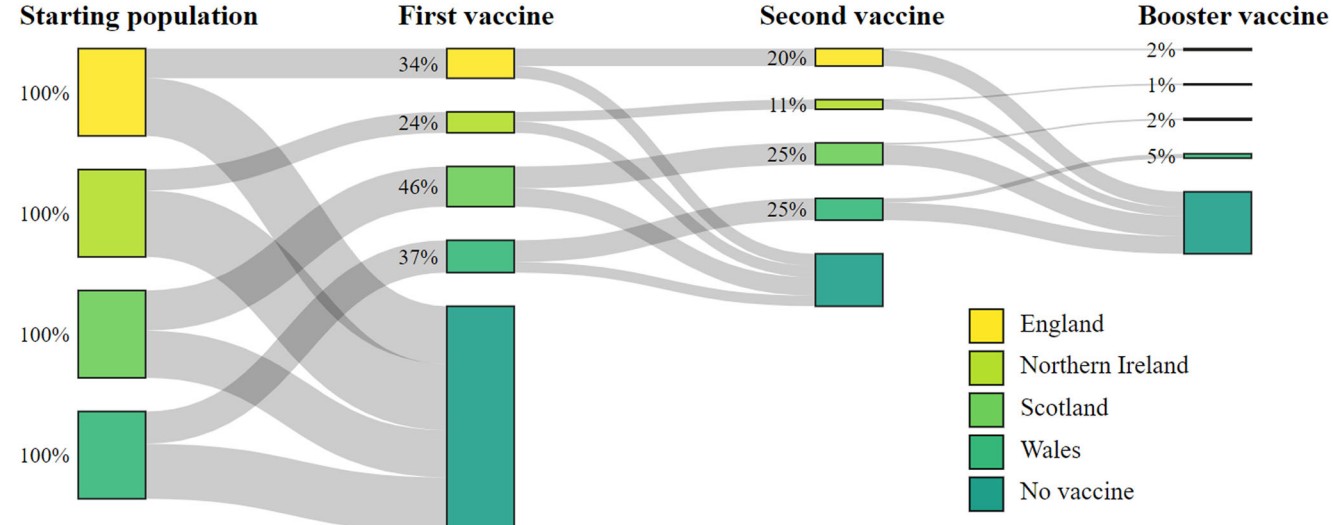

**Fig. 1 | Sankey diagram demonstrating vaccine uptake.** The proportional uptake of COVID-19 vaccine in CYP for each UK nation.

to receive the first (aHR 0.45, 95%CI 0.44–0.45) and second vaccination (aHR 0.3, 95%CI 0.29–0.31) compared to 16–17 year-olds (Table 2, Fig. 2). Uptake of the first vaccination was less likely for 5–11 year-olds (aHR 0.0013, 95%CI 0.0012-0.0014) compared to 16–17 year-olds, however, there was no difference in uptake of second vaccination (aHR 1.36, 95%CI 0.96-1.92). CYP in households of two were less likely to receive first (aHR 0.8, 95%CI 0.77-0.83) and second (aHR 0.87, 95%CI 0.82-0.92) vaccine compared to households of three. Households of four (aHR 1.17, 95%CI 1.14–1.19) and five or more (aHR 1.1, 95%CI 1.08–1.13) were more likely to receive the vaccine compared to households of three. CYP in unvaccinated households were 76% less likely (aHR 0.24, 95%CI 0.23-0.25) to receive the vaccine, and fully vaccinated households were twice as likely (aHR 2.33, 95%CI 2.29–2.37) to receive the first vaccine compared to partially vaccinated households (Table 2, Fig. 2).

In Scotland, males were less likely to receive their first (aHR 0.94, 95%CI 0.93–0.95), second (aHR 0.97, 95%CI 0.96–0.98) and booster vaccines (aHR 0.94, 95%CI 0.90-0.97) compared to females (Table 2,

Fig. 2). CYP aged 12–15 years old were less likely to receive their first vaccination (aHR 0.77, 95%CI 0.77-0.78), but more likely to receive their second vaccination (aHR 1.1, 95%CI 1.08-1.11) compared to 16–17 year-olds. CYP aged 5–11 years old were less likely to receive their first (aHR 0.17, 95%CI 0.17-0.17) and second (aHR 0.47, 95%CI 0.44–0.50) vaccination compared to 16–17 year-olds. Scotland showed a similar association with household size and likelihood of vaccine uptake to that of England (Table 2). CYP living in unvaccinated households were less likely to receive the first (aHR 0.29, 95% CI 0.27-0.3) and second (aHR 0.79, 95%CI 0.71-0.87) vaccine compared to partially vaccinated households, however there was no difference for booster doses (aHR 0.96, 95%CI 0.57-1.6). However, CYP in fully vaccinated households were more likely to receive first (aHR 1.58, 95%CI 1.55-1.61), second (aHR 3.79, 95%CI 3.66, 3.93) and booster doses (aHR 1.79, 95%CI 1.53-2.09) compared to partially vaccinated households.

Wales found a similar association to Northern Ireland in that males were less likely to receive their first (aHR 0.93, 95%CI 0.92–0.94) and

**Table 2 | Multistate model results for each nation : Adjusted hazard ratios and 95% confidence intervals for demographic variables for vaccine transitions resulting from the multistate model for individual nations**

| Country | Transition | Male | Age 12–15 | Age 5–11 | Household n = 2 | Household n = 4 | Household n ≥ 5 | Unvaccinated | Fully vaccinated |
|---|---|---|---|---|---|---|---|---|---|
| England | 1st dose | 0.97 (0.96, 0.97) (P = 0 ***) | 0.72 (0.72, 0.72) (P = 0 ***) | 0.11 (0.11, 0.11) (P = 0 ***) | 0.86 (0.85, 0.87) (P = 0 ***) | 1.11 (1.11, 1.12) (P = 0 ***) | 0.89 (0.89, 0.9) (P = 0 ***) | 0.11 (0.11, 0.11) (P = 0 ***) | 2.83 (2.8, 2.85) (P = 0 ***) |
| | 2nd dose | 0.98 (0.98, 0.99) (P = 0 ***) | 0.84 (0.84, 0.85) (P = 0 ***) | 0.8 (0.77, 0.84) (P = 0 ***) | 0.92 (0.91, 0.93) (P = 0 ***) | 1.05 (1.04, 1.06) (P = 0 ***) | 0.86 (0.86, 0.87) (P = 0 ***) | 0.59 (0.56, 0.61) (P = 0 ***) | 1.3 (1.29, 1.32) (P = 0 ***) |
| | Booster dose | 0.97 (0.95, 0.98) (P = 0 ***) | 0.06 (0.06, 0.07) (P = 0 ***) | 0 (0, Inf) (P = 0.8) | 0.91 (0.88, 0.95) (P = 0 ***) | 0.99 (0.97, 1.01) (P = 0.44) | 0.79 (0.77, 0.81) (P = 0 ***) | 0.62 (0.51, 0.74) (P = 0 ***) | 1.31 (1.26, 1.36) (P = 0 ***) |
| Northern Ireland | 1st dose | 0.87 (0.86, 0.88) (P = 0 ***) | 0.45 (0.44, 0.45) (P = 0 ***) | NA | 0.8 (0.77, 0.83) (P = 0 ***) | 1.17 (1.14, 1.19) (P = 0 ***) | 1.1 (1.08, 1.13) (P = 0 ***) | 0.24 (0.23, 0.25) (P = 0 ***) | 2.33 (2.29, 2.37) (P = 0 ***) |
| | 2nd dose | 0.9 (0.88, 0.92) (P = 0 ***) | 0.3 (0.29, 0.31) (P = 0 ***) | NA | 0.87 (0.82, 0.92) (P = 0 ***) | 1.11 (1.07, 1.14) (P = 0 ***) | 0.97 (0.94, 1.01) (P = 0.1) | 0.57 (0.53, 0.61) (P = 0 ***) | 1.5 (1.46, 1.54) (P = 0 ***) |
| | Booster dose | 0.95 (0.88, 1.02) (P = 0.17) | 0.16 (0.14, 0.18) (P = 0 ***) | NA | 1.03 (0.84, 1.25) (P = 0.8) | 0.99 (0.89, 1.11) (P = 0.93) | 0.89 (0.8, 0.99) (P = 0.03 **) | 0.62 (0.44, 0.88) (P = 0.01 **) | 1.37 (1.24, 1.52) (P = 0 ***) |
| Scotland | 1st dose | 0.94 (0.93, 0.95) (P = 0 ***) | 0.77 (0.77, 0.78) (P = 0 ***) | 0.17 (0.17, 0.17) (P = 0 ***) | 0.97 (0.96, 0.99) (P = 0 ***) | 1.07 (1.06, 1.08) (P = 0 ***) | 0.85 (0.84, 0.85) (P = 0 ***) | 0.29 (0.27, 0.3) (P = 0 ***) | 1.58 (1.55, 1.61) (P = 0 ***) |
| | 2nd dose | 0.97 (0.96, 0.98) (P = 0 ***) | 1.1 (1.08, 1.11) (P = 0 ***) | 0.47 (0.44, 0.5) (P = 0 ***) | 0.98 (0.96, 1) (P = 0.04 **) | 1.05 (1.04, 1.07) (P = 0 ***) | 0.9 (0.89, 0.92) (P = 0 ***) | 0.79 (0.71, 0.87) (P = 0 ***) | 3.79 (3.66, 3.93) (P = 0 ***) |
| | Booster dose | 0.94 (0.9, 0.97) (P = 0 ***) | 0.07 (0.06, 0.07) (P = 0 ***) | 0 (0, Inf) (P = 0.93) | 1.02 (0.96, 1.08) (P = 0.49) | 1.01 (0.97, 1.06) (P = 0.59) | 0.77 (0.73, 0.81) (P = 0 ***) | 0.96 (0.57, 1.6) (P = 0.86) | 1.79 (1.53, 2.09) (P = 0 ***) |
| Wales | 1st dose | 0.93 (0.92, 0.94) (P = 0 ***) | 0.46 (0.45, 0.46) (P = 0 ***) | 0.06 (0.06, 0.06) (P = 0 ***) | 0.83 (0.81, 0.86) (P = 0 ***) | 1.14 (1.12, 1.16) (P = 0 ***) | 0.99 (0.98, 1.01) (P = 0.25) | 0.19 (0.18, 0.2) (P = 0 ***) | 2.45 (2.41, 2.49) (P = 0 ***) |
| | 2nd dose | 0.97 (0.95, 0.98) (P = 0 ***) | 0.96 (0.95, 0.97) (P = 0 ***) | 0.42 (0.38, 0.47) (P = 0 ***) | 0.94 (0.91, 0.97) (P = 0 ***) | 1.05 (1.03, 1.07) (P = 0 ***) | 0.93 (0.91, 0.95) (P = 0 ***) | 0.64 (0.6, 0.68) (P = 0 ***) | 1.3 (1.28, 1.33) (P = 0 ***) |
| | Booster dose | 1.02 (0.99, 1.05) (P = 0.26) | 0.14 (0.14, 0.15) (P = 0 ***) | NA (NA, NA) (P = NA) | 0.89 (0.83, 0.95) (P = 0 ***) | 0.97 (0.93, 1.01) (P = 0.1) | 0.82 (0.79, 0.86) (P = 0 ***) | 0.5 (0.41, 0.6) (P = 0 ***) | 1.38 (1.31, 1.45) (P = 0 ***) |

The reference for each group is female, aged 16–17 years old, a household occupancy of 3, and a partially vaccinated household.
*Asterisk indicates significance of two tailed P-values, **P < 0.05, ***P < 0.01.

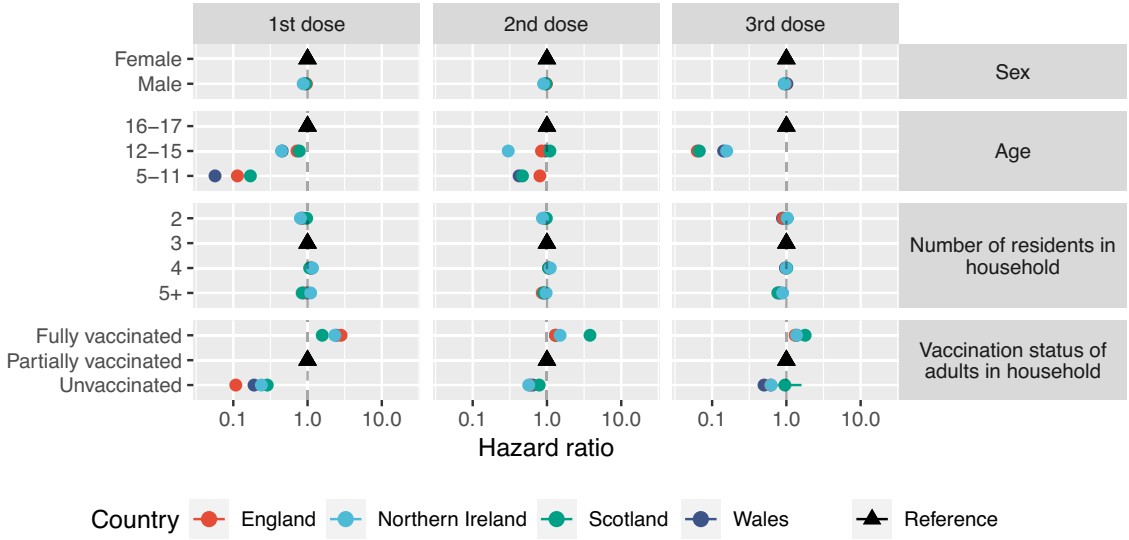

**Fig. 2 | Multistate model results.** Adjusted hazard ratio coefficient and 95% confidence intervals resulting from the multistate model for each transition for all four UK countries. Black triangles indicate reference groups. aHR for Northern Ireland's 5–11 year-olds receiving their first dose is 0 and sits beyond the scale presented. The values for the Adjusted hazard ratio coefficients and 95% confidence interval are presented in Table 2.

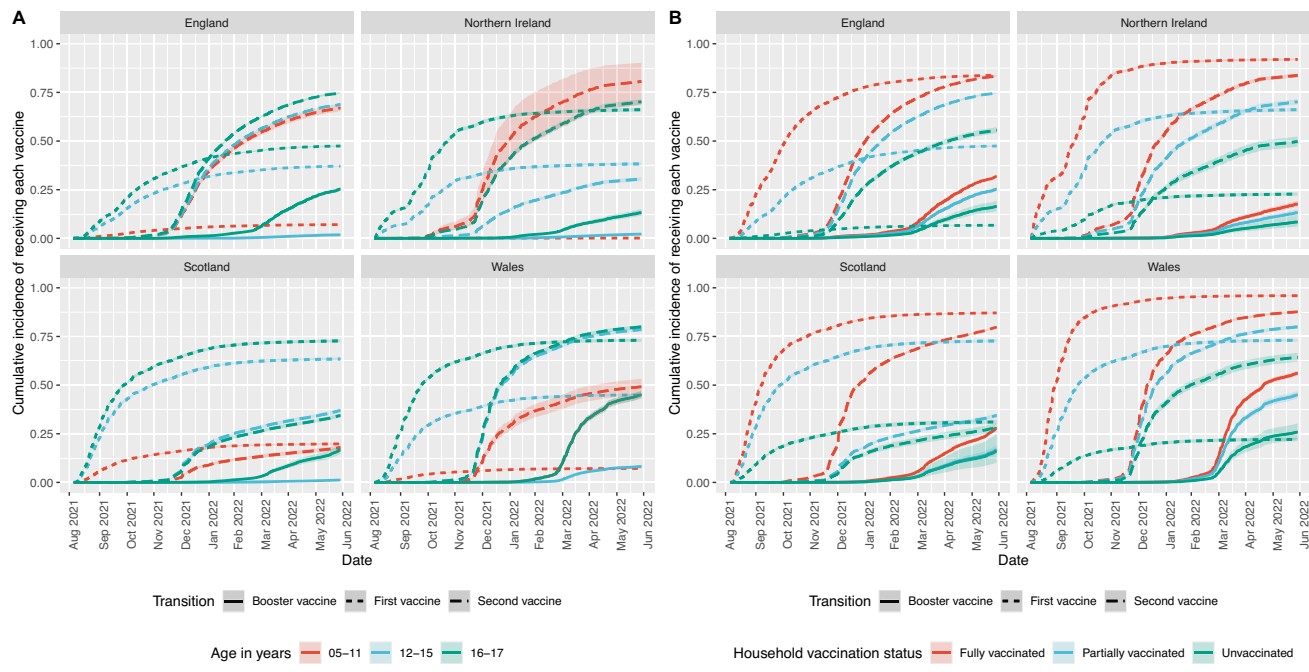

**Fig. 3 | Cumulative incidence of vaccine uptake.** The estimated cumulative incidence and 95% confidence intervals for vaccine uptake in each country for each vaccine in (**A**) different age groups and (**B**) household vaccination status.

second (aHR 0.97, 95%CI 0.95–0.98) vaccination compared to females, but there was no difference between males and females receiving their booster dose (aHR 1.02, 95%CI 0.99–1.05). Similarly, to England, CYP in Wales aged 12–15 years old (aHR 0.46, 95%CI 0.45, 0.46) and 5–11 years old (aHR 0.06, 95%CI 0.06–0.06) were less likely to receive their first vaccination compared to 16–17 year-olds (Table 2, Fig. 2). A similar association was found for second vaccination and booster doses (Table 2, Fig. 2). CYP in households of two were less likely to receive their first vaccine (aHR 0.83, 95%CI 0.81–0.86), whilst households of four were more likely to receive the first vaccine (aHR 1.14, 95%CI 1.12–1.16) compared to households of three. In Wales, there was insufficient difference in vaccine uptake in household of 5 or more

(aHR 0.99, 95%CI 0.98-1.01) compared to households of three. Wales showed a similar association with household vaccination status and likelihood of vaccine uptake to that of England and Northern Ireland (Table 2, Fig. 2).

Cumulative incidence plots for sex and number of residents in the household are provided in Supplementary Fig. 2.

### UK meta-analysis
A meta-analysis combining the results from all four nations showed that vaccine uptake across the UK was affected by age, sex, number of residents in the household, and household vaccination status (Fig. 4, Table 3).

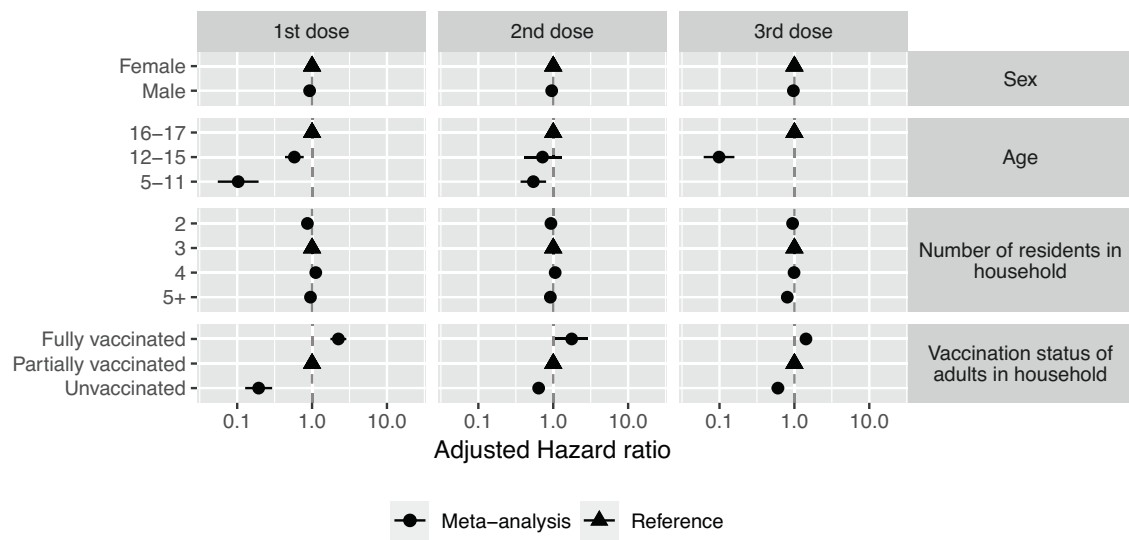

**Fig. 4 | Results of the meta-analysis.** Pooled adjusted hazard ratio coefficients and 95% confidence intervals obtained from the random effects meta-analysis model, synthesising results from England, Northern Ireland, Scotland and Wales. Triangles correspond to reference variables. Northern Ireland's 5–11 year-olds were removed from this analysis due to small numbers. The values for the pooled Adjusted hazard ratio coefficients and 95% confidence interval are presented in Table 3.

**Table 3 | Meta-analysis results: Pooled aHR and 95% confidence intervals for each variable and vaccine across all four nations**

| Variable | Type | Vaccine | aHR | 95%CI lower | 95%CI upper | Between study variance ($\tau^2$) |
|---|---|---|---|---|---|---|
| Age | 5–11 years old | 1st dose | 0.10 | 0.06 | 0.19 | 0.31 |
| | | 2nd dose | 0.54 | 0.37 | 0.80 | 0.12 |
| | 12–15 years old | 1st dose | 0.58 | 0.43 | 0.77 | 0.09 |
| | | 2nd dose | 0.72 | 0.40 | 1.28 | 0.35 |
| | | Booster dose | 0.10 | 0.06 | 0.16 | 0.23 |
| Sex | Male | 1st dose | 0.93 | 0.89 | 0.97 | 0.00 |
| | | 2nd dose | 0.95 | 0.92 | 0.99 | 0.00 |
| | | Booster dose | 0.97 | 0.93 | 1.01 | 0.00 |
| Number of residents in household | 2 | 1st dose | 0.87 | 0.80 | 0.94 | 0.01 |
| | | 2nd dose | 0.93 | 0.89 | 0.97 | 0.00 |
| | | Booster dose | 0.95 | 0.88 | 1.02 | 0.00 |
| | 4 | 1st dose | 1.12 | 1.08 | 1.16 | 0.00 |
| | | 2nd dose | 1.06 | 1.04 | 1.08 | 0.00 |
| | | Booster dose | 0.99 | 0.97 | 1.01 | 0.00 |
| | 5+ | 1st dose | 0.95 | 0.85 | 1.07 | 0.01 |
| | | 2nd dose | 0.92 | 0.87 | 0.96 | 0.00 |
| | | Booster dose | 0.80 | 0.77 | 0.84 | 0.00 |
| Vaccination status of adults in household | Unvaccinated | 1st dose | 0.19 | 0.13 | 0.29 | 0.18 |
| | | 2nd dose | 0.64 | 0.56 | 0.73 | 0.02 |
| | | Booster dose | 0.60 | 0.50 | 0.73 | 0.02 |
| | Fully vaccinated | 1st dose | 2.25 | 1.76 | 2.87 | 0.06 |
| | | 2nd dose | 1.76 | 1.06 | 2.92 | 0.26 |
| | | Booster dose | 1.43 | 1.27 | 1.61 | 0.01 |

Northern Ireland's results for 5–11 year-olds were excluded from the meta-analysis due to small numbers.

Compared to 16–17 year-olds, CYP aged 12–15 were 42% less likely to receive the first vaccine (aHR 0.58, 95%CI 0.43–0.77), and 90% less likely to receive the booster vaccine (aHR 0.10, 95%CI 0.06–0.19) having adjusted for sex, number of residents in the household and household vaccination status. However, there was insufficient evidence to detect a difference for second vaccine (aHR 0.72, 95%CI 0.40–1.28). CYP aged 5–11 years old were 90% less likely to receive first vaccine (aHR 0.10, 95%CI 0.06–0.19) and 46% less likely (aHR 0.54, 95% CI 0.37–0.80) to receive their second vaccine compared to 16–17 year-olds having adjusted for all other factors (Table 3). Males were 7% less likely (aHR 0.93, 95%CI 0.89–0.97) to receive their first vaccine compared to females and 5% less likely (aHR 0.95, 95%CI 0.92–0.99) to receive their second vaccine, having adjusted for all other factors. CYP in a household of two were 11% (aHR 0.89, 95%CI 0.80–0.94) and 7%

(aHR 0.93, 95%CI 0.89–0.97) less likely to receive their first and second doses compared to households of three residents, respectively. Households with five or more residents showed no difference in uptake of their first vaccine compared to households of 3 residents but were 8% (aHR 0.92, 95%CI 0.87–0.96) and 20% (aHR 0.80, 95%CI 0.77–0.84) less likely to receive their second or booster dose, respectively. In contrast, CYP in a household of four people were more likely to receive their first (aHR 1.12, 95%CI 1.08–1.16) and second vaccine (aHR 1.06, 95%CI 1.04–1.08) compared to households of 3 residents.

Household vaccination status was an important factor in the uptake of all three vaccines, having adjusted for differences in age, sex, and number of residents in the household. CYP residing in a household with fully vaccinated adults were 2.25 times as likely to receive their first vaccine (aHR 2.25, 95%CI 1.76–2.87), 1.76 times as likely to receive their second vaccine (aHR 1.76, 95%CI 1.06–2.92), and 1.43 times as likely to receive their booster vaccine (aHR 1.43, 95%CI 1.27–1.61) compared to an individual in a partially vaccinated household, holding all other factors constant. Those in an unvaccinated household were 81% less likely to receive their first vaccine (aHR 0.19, 95%CI 0.13–0.29), 36% less likely to receive their second vaccine (aHR 0.64, 95%CI 1.56–0.73), and 40% less likely to receive their booster dose (aHR 0.60, 95%CI 0.50–0.73), compared to CYP in partially vaccinated households.

A large between cohort variability was observed in the vaccine uptake for 5–11 year-olds (Supplementary Table 2), resulting from the very small numbers of CYP aged 5–11 years old in Northern Ireland receiving their vaccine, so this group was removed from the analysis in the form of a sensitivity analysis. Excluding Northern Ireland's 5–11 year-olds showed that despite the variability, the results remained robust (see Supplementary Table 3).

## Discussion

In this analysis of prospective cohorts in England, Northern Ireland, Scotland, and Wales, we found that approximately one-third of CYP aged 5–17 years received their first COVID-19 vaccine uptake by 31st May 2022. In general, uptake was lower for second and booster vaccines in all circumstances. The UK-wide meta-analysis indicates that age, sex, household composition and vaccination status of adults living in the same household were important factors in vaccine uptake in CYP.

Vaccine uptake was lower in 12–15 year-olds, and lower still in 5–11 year-olds compared to uptake in 16–17 year-olds. The integration of staggered start dates in the multi-state models to capture the rollout of vaccines for each age group accounts for the delayed vaccine schedule that was experienced by 12–15 and 5–11 year-olds. Despite this, the findings indicated that both the uptake and rate of uptake was lower in these age groups (12–15 years 1st vaccine: aHR 0.58, 95%CI 0.43–0.77, 5–11 year-olds 1st vaccine: aHR 0.10, 95%CI 0.06–0.19, Fig. 3). The disparity in uptake was less severe for the second (12–15 years 2nd vaccine aHR 0.72, 95%CI 0.40–1.28, 5–11 year-olds 2nd vaccine: aHR 0.54, 95%CI 0.37–0.80) and booster doses (12–15 years 1st vaccine aHR 0.10, 95%CI 0.06–0.16). These findings are in line with research investigating intentions regarding COVID-19 vaccination of CYP from both CYP themselves and parents in the UK and globally, which have also shown that vaccine hesitancy increased inversely with age[19,26,27]. There is a widespread belief that the vaccines are not as safe for CYP as they are for adults, with research from interviews in the UK, Ireland, and the USA indicating that some parents and young people are sceptical that the risk of vaccination outweighs the benefit[28–31]. Younger age groups are associated with a decrease in COVID-19 morbidity and mortality, and the perception that the vaccine is less important for CYP is a possible reason for a reduction in uptake. Caregivers will play a vital role in the vaccination of these age groups, as those under the age of 16 years will require consent from

their legal guardian, and those aged 16–17 who are able to give their own consent are still likely to be influenced by their guardians. The cumulative incidence of vaccine uptake revealed the longitudinal association between age and uptake for each nation over time. The rate of uptake followed a very similar patterns across these groups. It also showed that despite being eligible for less time during the study period, uptake in younger age groups also plateaued for the first vaccine.

The vaccination status of adults living in the same household was also an important determinant of vaccine uptake. CYP living in fully vaccinated households were twice as likely (aHR 2.2.5, 95%CI 1.76–2.87) to receive their first vaccine compared to those in partially vaccinated households. An overall majority (96%) of CYP residing in households with unvaccinated adults were also unvaccinated, however this proportion reduced for the second and third doses. An unvaccinated household could indicate limited access to vaccines and/or a decision-based barrier, representing adults who have opted out of the vaccine rollout. One or more of the adults residing in the same household are likely to be a caregiver or in some sort of care-giving role to the CYP and will be responsible for providing legal consent for those aged under 16 years. Additionally, parents and guardians are likely to have a strong influence over a CYP's knowledge and opinion on COVID-19 vaccinations[32]. A guardian or parent may have to provide logistical means for a CYP to access a vaccine, providing a further barrier if they hold opposing opinions on COVID-19 vaccines.

There were small differences in uptake regarding sex and the number of residents in the household. Males were marginally less likely to receive vaccines compared to females. This is in line with research investigating uptake in adult groups which has also tended to show a slightly higher uptake in females compared to males[33]. Households with two or five or more people were less likely to receive their vaccine, while households with four individuals were more likely to receive their vaccine compared to households of three. A household of two implies a single adult living with a child, giving this adult greater responsibility over influence and access regarding the COVID-19 vaccine. Larger households could indicate other factors, such as households with many children, multiple generations or houses of multiple occupancy (HMO). Further research could work to identify risk factors in household types to investigate more direct associations, which may be influenced by factors such as living with vulnerable people, familial structure or deprivation.

An additional potential factor that may have influenced vaccine uptake was the underlying COVID-19 morbidity and mortality in CYP and/or the general population. It is known that COVID-19 morbidity and mortality varied with different dominant variants[24]. During this study period in the UK, the Delta and Omicron variants were dominant. However, the Delta variant was only dominant for a maximum of 4.5 months from the beginning of the vaccine roll out programme. For this reason, it was not possible to investigate the impact of dominant variant on vaccine uptake owing to low event rates. For 5–11 year-olds for example, the vaccine roll out period was only applicable during the Omicron dominant period.

There appeared to be heterogeneity in vaccine uptake, particularly with respect to age-groups and household vaccination status, between nations. Uptake in younger age groups (5–11 and 12–15 years) were particularly low in Northern Ireland, despite Northern Ireland being the first nation to approve the vaccine rollout to these age groups. In contrast, Scotland showed an increased uptake for younger age groups overall. Uptake in 5–11 year-olds is the most extreme of these cases, where aHRs ranged from 0.0013 (Northern Ireland) to 0.17 (Scotland) for the first vaccine. Similarly, for household vaccination status, the aHR for unvaccinated households ranged from 0.11 (England) to 0.29 (Scotland) for first vaccine. Further investigation into differences in national vaccination strategies of CYP could shed light on the reason for this heterogeneity in uptake.

This is the first study to use pooled data from all four UK nations to assess vaccine uptake in CYP. All nations used common definitions and methods, which improved the consistency of the pooled meta-analysis results. It is also the first application of a multi-state model to be used on COVID-19 vaccination uptake, allowing us to account for competing events such as COVID-19 infection, which delays vaccination by 28 days, and death. By assigning a universal start date to all nations defined by the JCVI recommendation, we have excluded many individuals who would have received their vaccines early based on the assumption that these individuals were more likely to have underlying health conditions and/or reside in clinically vulnerable households. Future work could explore vaccine uptake in CYP with alternative vaccine roll-out strategies.

This study was somewhat limited by the age data available for CYP. Age was assigned at the start of the cohort study, 4th August 2021, for England, Northern Ireland and Wales, while Scotland defined their ages 1st March 2022. Wales was able to update CYP ages at the start of each new vaccine rollout to capture those that had birthdays and were moved up in the vaccine schedule. We were also limited by the reliance on the accuracy of COVID-19 test data to confirm cases of COVID-19 infection. Confirmatory Reverse Transcriptase Polymerase Chain Reaction (RT-PCR) testing was still in place for the duration of this study. The extent to which RT-PCR tests were used is unknown, particularly for CYP whose uptake of these tests may have been affected by the practicality of using them. The rate of reporting for positive lateral flow devices was also unknown. However, it remains important to account for competing risks when known, even in the presence of potentially missing data. A further limitation of the study is the inability to account for household vaccination status as a time-varying covariate; owing to the scale and nature of the data, it would be computationally impractical to run such analyses across all four nations.

Recent data from the Office for National Statistics (ONS) ([34], accessed 27th January 2023) shows that coverage in England for CYP aged 5–17 is similar to the coverage we obtained by the conclusion of our study: 30%, 23%, and 2% for the first, second, and booster doses, respectively. It may be that CYP are not currently a priority, as the UK's governments choose to focus resources on additional booster doses for adults. Should this strategy change so that CYP becomes a priority, additional measures may be needed to improve vaccine uptake in younger CYP and associated parental/guardian consent.

Vaccine uptake is a key component in reducing the effect of COVID-19 on the population, and uptake in CYP will play its own role. Additionally, there is evidence emerging that COVID-19 vaccination reduces the likelihood of MICS-C, providing a further incentive to vaccinate this group[35]. The results of this research can be used to optimise future rollouts by targeting CYP that are being under-represented in the vaccination programmes. Changes to policy, methodology, or the messaging used to approach these groups could result in better uptake, however care needs to be taken to address the concerns of legal guardians, as well as the CYP themselves. Our results suggest that targeting younger CYP and those who reside in an unvaccinated household will improve uptake. There may be ethical considerations regarding consent when targeting vaccine hesitant households. Vaccine hesitancy can result from a combination of confidence—a person's trust in the effectiveness, safety, and honesty of the vaccine, health professionals, and policymakers; complacency—a person's sense of urgency or necessity regarding the vaccine; or convenience—the physical, logistical and financial barriers associated with receiving the vaccine[36]. If a household is unvaccinated through choice (confidence), then improved education around the vaccines could increase uptake, however if a household is unvaccinated as a result of poor access, then a boost in this access (convenience) will likely increase vaccine uptake. Coordination with schools would be pivotal in overcoming barriers faced by these CYP.

School coordination should provide repeat opportunities to CYP for vaccination to capture those that have or have recently had a COVID-19 infection. Unvaccinated households represent only 10% of the cohort, so strategies to increase uptake in these populations may be less effective than prioritising age. Further work to identify root causes for poor vaccine uptake in CYP could assist with identifying target groups more effectively.

## Methods

### Study design

Each of the four UK nations constructed cohorts using individual-level, linked, anonymised electronic health record (EHR) data sources and administrative data sources available from their respective national data sources and trusted research environments (TREs). These data sources included clinical and demographic characteristics, vaccine-related information, and SARS-CoV-2 infection details obtained from Reverse Transcriptase Polymerase Chain Reaction (RT-PCR) and Lateral Flow Test (LFTs).

Our study period ran from 4th August 2021 to 31st May 2022. The JCVI approval for 16–17 year-olds determined the start date for the study. Information was collected from the 7th July 2021, 28 days prior to the study start date, enabling identification of individuals who would be entering the study infected with COVID-19, and how far through an infection they were. Additionally, we performed a sensitivity analysis in Northern Ireland, Scotland and Wales with a study end date of 31st December 2022 to identify potential changes in vaccine uptake.

Each cohort consisted of individuals aged 5–17 during the study period. Wales and England characterised an individual's age at the start of a relevant vaccine rollout. Northern Ireland assigned age groups at the start of the study period, and Scotland assigned age groups from March 2022. Vaccine information was limited to the first, second, and booster doses with BNT162b2 (tozinameran; Pfizer–BioNTech), ChAdOx1 nCoV-19 (Oxford–AstraZeneca), and mRNA-1273 (Elasomeran; Moderna) vaccines with a minimum of 28 days between doses. All individuals were followed up from the start of the study period to cohort end (31st May 2022), the date they moved out of the nation or until death. Patients hospitalised for >1 week during the study period may have experienced a disruption to their vaccine schedule, but the magnitude of a potential delay to their vaccination schedule is unclear for these individuals. For this reason, we excluded these individuals (<0.1% of the total cohort) from the study.

### Study data sources

For England, data were accessed through the Oxford-Royal College of General Practitioners (RCGP) Research and Surveillance Centre (RSC) database covering >19 million (around 33% of the English population) individuals. This near real-time dataset is representative of the national population[37] and links individual patient-level primary care data with the National Immunisation Management Service (NIMS) for vaccine uptake, Hospital Episode Statistics for hospitalisation and intensive care unit admissions, and Office for National Statistics (ONS) data for certificated cause of death. Pseudonymisation was conducted using a National Health Service (NHS) digital-approved process, allowing pseudonymised NHS numbers (unique national IDs) to link individual patient-level data to other datasets to supplement primary care data.

For Northern Ireland, data were accessed through the Honest Broker Service (HBS). Vaccination data from the Vaccine Management System were linked using an anonymised study identifier that replaced each individual's unique health and care number to the National Health Application and Infrastructure Services (NHAIS) system, which contains information on all patients registered with a primary care physician construct the cohort, covering 1.9 million individuals (entire population) and COVID-19 testing data from the Northern Ireland central testing register.

For Scotland, data were accessed through the Early Pandemic Evaluation and Enhanced Surveillance of COVID-19 (EAVE II) platform[38], which contains electronic health records for 5.4 million (~99% of the population) individuals in Scotland. We had data from all 940 Scottish primary care practices. These were linked via the Community Health Index (CHI) to the Electronic Communication of Surveillance in Scotland (ECOSS; national database for all virology testing including NHS and UK Government test centre data) and National Records Scotland (death certification) data as part of the EAVE II platform.

For Wales, data were accessed through the Secure Anonymised Information Linkage (SAIL) Databank, a TRE dedicated to storing and linking several health, well-being services, and administrative data[39]. Information for approximately half a million CYP were taken from primary care General Practitioner (GP) records, secondary hospital data, and various administrative sources, representing ~85% of the population of Wales.

### Outcomes

Our main outcomes were time to COVID-19 vaccinations under the primary vaccination schedule and the booster vaccination programme while accounting for competing events of SARS-CoV-2 infection and death. Individuals were followed up from the date of eligibility for their first primary dose, which was based on their age at entry into the cohort.

### Population characteristics and covariates

Several factors have been shown to affect vaccine uptake in UK populations[33,40]. To enable a comprehensive meta-analysis, we were limited to factors available to all four UK nations. Four factors were selected for investigation: age, sex, vaccination status of the adults in the household, and the number of people in the same household. The number of people in the household (household $n$) is a count of the number of individuals (both adults and children) living at the same address, with categories grouped as 2, 3, 4, and 5 +. The household vaccination status was classified as fully vaccinated if all adults in the same household had received their first vaccine, partially vaccinated if at least one, but not all the adults had received their first vaccine, and unvaccinated if no adults in the household had received their first vaccination.

Age was grouped according to the vaccine schedule groupings: 16–17, 12–15, and 5–11 years old and were defined at the start of

eligibility for the following age range (e.g., if they turned 16 after the 16–17 rollout, but before the 12–15 vaccine rollout, they were included in the 16–17 age-group at the start of their eligibility). If this resolution of age data was unavailable, the age on a specified date was used.

We include an extended study period and additional covariates that have been shown to influence vaccine uptake in other studies[17–22], including ethnicity, deprivation and urbanicity, in the form of a sensitivity analysis where possible, as these were not available for all regions.

### Statistical analysis

Standard survival methods, such as the Cox proportional hazard method, have been used to model the uptake of vaccines[33,41]. However, due to government advice to avoid vaccination for 28 days following infection, this makes an individual ineligible for vaccination following COVID-19 and introduces complexity to the model. One possible method to account for this complexity was to use a multi-state model. This method was designed to incorporate intermediate states into the Cox proportional hazard model to appropriately account for competing risks, including COVID-19 infection and death.

Multi-state models have been commonly used to model complex disease progression with intermediate states between the entry state and the absorbing state (usually death) (e.g.[42,43],). However, they have also been used to model influenza vaccine uptake[44]. This method was particularly well suited to answer our research question by accounting for those ineligible for their vaccine during and following infection.

Each nation performed a bi-directional multi-state model using Cox proportional hazards regression to estimate the transition between states, accounting for competing events (infection and death). This bidirectional model allowed CYP to transition to and from a state of infection and subsequent vaccinations and death (Fig. 5). The assumption of proportional hazards was assessed through visual inspection of the Schoenfeld residual plots. The reference group was defined as CYP who were female, aged 16–17 years old and lived in a partially vaccinated household of three residents.

The multi-state model consisted of 6 states: Unvaccinated, COVID-19 infection, Primary vaccination, Secondary vaccination, Third/Booster vaccination and Death (Fig. 5). Individuals could enter the model either as an eligible and unvaccinated candidate, or as an infected and ineligible candidate if they had an infection up to 28 days prior to the eligibility date. This ensured that individuals that were not eligible to receive vaccination owing to COVID-19 infection (i.e.,

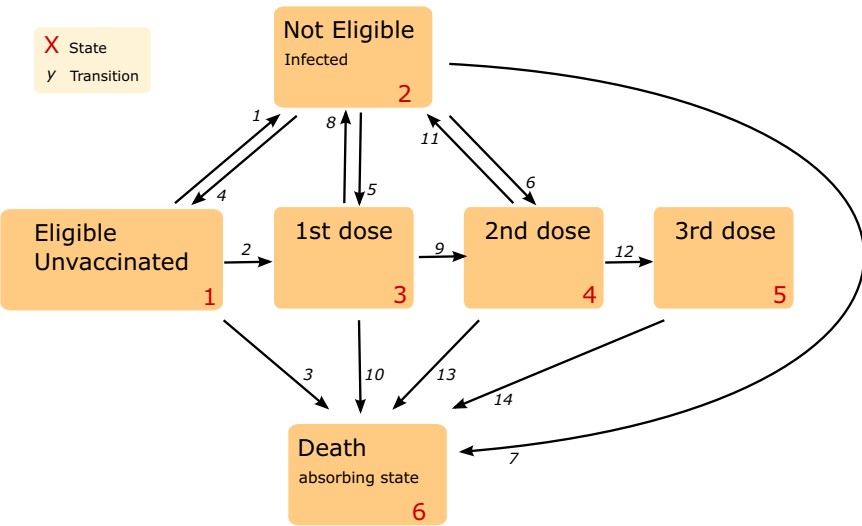

**Fig. 5 | The proposed multi-state model.** This model consists of 6 states and 14 possible transitions between them. Infection following a booster dose is not of interest and has been excluded from the model. Death acts as an absorbing state.

infection was a competing risk for vaccine upon study entry) were adjusted for in the model. Each person could move up the vaccination schedule, or into the infection state where they will remain for 28 days in line with government recommendations before returning to the state they had been in prior to infection, or into the absorbing death state. CYP could also be considered ineligible following vaccination, as there is a predefined delay between each vaccine. We chose not to include this in the multi-state model as this state of ineligibility would behave uniformly to all who received a vaccine. Individuals were censored at the first of study end, death, or the date of change in national residency. There were 14 possible transitions between the states, including bidirectionality between the infection and eligible/ vaccinated states. If a person received their vaccination before the 28 days were complete (i.e., they were still in the ineligible state), they were automatically moved to the relevant eligible state before moving to the next state. The key transitions for this study were those that represent the transition from an eligible state to their first, second or booster dose. Cumulative incidence of first, second and booster dose were calculated from the multistate models, having adjusted for infection and mortality as competing risks.

The derived aggregate data from each nation (reported as adjusted log hazard ratios (aHR) and corresponding standard errors) were exported from the respective TREs and meta-analysed in a two-stage individual participant data random effects meta-analysis[45]. A pooled analysis could not be performed with the data available given the sovereign nature of the datasets used. It was not possible to share large, linked, individual level electronic health records across jurisdictions. In keeping with related, previous analyses (e.g[46,47].) we therefore chose to perform a meta-analysis, pooling the data across regions using the extracted coefficients. Effect estimates were presented as hazard ratios and corresponding 95% confidence intervals. Restricted maximum likelihood estimation was used to estimate the pooled coefficients. Heterogeneity between the estimated effects between the nations were quantified using the between-study variance ($\tau^2$). $I$-squared was used to assess the proportion of total variability owing to between-study heterogeneity.

In addition, we assessed the robustness of the results obtained from the UK-wide meta-analysis using a sensitivity analysis excluding estimates for 5–11 year-olds in Northern Ireland owing to very small numbers of vaccine uptake in this age-group. National level data analysis scripts were created using Rstudio v4.1.3 for running within the TREs. Meta-analsysis scripts were created and performed using Rstudio v4.2.0. Analytical R scripts made use of using the *Survival, Mstate* and *Metafor* packages[48–50].

### Ethics and permissions

In England, ethical approval was granted by the Health Research Authority London Central Research Ethics Committee (reference number REC reference 21/HRA/2786; integrated research application system number 30174). In Northern Ireland, study approval was granted by the Honest Broker Service (HBS) Governance Board (project number 064; the HBS process does not require separate National Research Ethics Service governance approval). In Scotland, ethical approval was granted by the National Research Ethics Service Committee (Southeast Scotland 02; reference number 12/SS/0201), and the approval for data linkage was granted by the Public Benefit and Privacy Panel for Health and Social Care (reference number 1920–0279). In Wales, research conducted within the Secure Anonymised Information Linkage Data-bank was done with the permission and approval of the independent Information Governance Review Panel (project number 0911). Individual written patient consent was not required for this study.

### Reporting summary

Further information on research design is available in the Nature Portfolio Reporting Summary linked to this article.

## Data availability

Aggregate data on COVID-19 vaccinations of CYP in the UK are provided in this publication's supplementary material. The patient-level data underlying this article provided by the RCGP and RSC database, NHAIS, EAVE II platform and the SAIL Databank cannot be shared publicly due to data protection and confidentiality requirements. Data can be made available to approved researchers for analysis after securing relevant permissions from the data holders via the relevant approval pathways.

## Code availability

The codes used for regional analysis and the combined meta analysis are available at https://github.com/HDRUK/DaCVaP/tree/main/DaCVaP2/Children-and-Young-People.

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

## Acknowledgements

This research is part of the Data and Connectivity National Core Study, led by Health Data Research (HDR) UK in partnership with the Office for National Statistics and funded by UK Research and Innovation (grant ref MC_PC_20058) [UA, SM, AA, SB, MJ, EL, SL, CR, DB, RAL, AS]. Data and Connectivity: COVID-19 Vaccines Pharmacovigilance National Core Study - Uptake, safety and effectiveness of COVID-19 vaccines in: pregnancy; children and young people; those receiving booster doses; and disease caused by different variants (2021.0158) [UA, SM, AA, SB, MJ, SL, LP, CR, DB, RAL, AS, RKO] is a partnership between University of Edinburgh, University of Oxford, University of Strathclyde, Queen's University Belfast, Swansea University, Imperial College London and the Office for National Statistics. This partnership was funded by HDR UK and The Alan Turing Institute. RKO is supported by the Academy of Medical Sciences/the Wellcome Trust/ the Government Department of Business, Energy and Industrial Strategy/the British Heart Foundation/ Diabetes UK Springboard Award (SBF006\1122). This work was supported by HDR UK, which receives its funding from HDR UK Ltd (HDR-9006) [SA, AA, SB, RGr, EL, RAL, RKO], funded by the UK Medical Research Council, Engineering and Physical Sciences Research Council, Economic and Social Research Council, Department of Health and Social Care (England), Chief Scientist Office of the Scottish Government Health and Social Care Directorates, Health and Social Care Research and Development Division (Welsh Government), Public Health Agency (Northern Ireland), British Heart Foundation (BHF) and the Wellcome Trust. This work was supported by the Administrative Data Research (ADR) Wales programme of work. ADR Wales, part of the ADR UK investment, unites research expertise from Swansea University Medical School and WISERD (Wales Institute of Social and Economic Research and Data) at Cardiff University with analysts from Welsh Government. ADR UK is funded by the Economic and Social Research Council (ESRC), part of UK Research and Innovation. This research was supported by ESRC funding, including Administrative Data Research Wales (ES/W012227/1) [AA, SB, RAL]. Additionally the authors acknowledge the support of BREATHE—The Health Data Research Hub for Respiratory Health (MC_PC_19004) [AS], which is funded through the UK Research and Innovation Industrial Strategy Challenge Fund and delivered through Health Data Research UK. This work was also supported by the Con-COV team funded by the Medical Research Council (grant number: MR/V028367/1) [AA, RAL].

## Author contributions

S.J.A. led the preparation and final editing of the manuscript. S.J.A. and R.K.O. designed the analysis. S.J.A. and R.Gr. led the data preparation and cleaning in Wales. S.J.A. performed the survival analysis for Wales. S.M. and D.T.B. accessed and verified the underlying data, and in addition to L.P. were responsible for data cleaning and preparation in Northern Ireland. S.M. performed the survival analysis for Northern Ireland. For England, U.A. and R.Go. accessed and verified the underlying data. In addition, R.Go. was responsible for data cleaning and U.A. for survival analysis. T.M. performed the data preparation and cleaning and the survival analysis for Scotland. C.R. accessed and verified the underlying data. S.J.A. led the meta-analysis of all UK nations. R.K.O. oversaw the analyses. S.J.A., U.A., S.M., T.M., A.A., F.A., S.N.A., S.B., R.Go., R.Gr., M.J., E.L., S.d.L., L.P., C.R., I.R., D.T.B., R.A.L., A.S., R.K.O. contributed to the conceptualisation of the study, drafting the paper, and revising the manuscript for important intellectual content. All authors have seen and approved the final text and the version to be published. A.S. was the study PI and oversaw all aspects of the work.

## Competing interests

AS has served on a number of UK and Scottish government COVID-19 advisory bodies. CR has served on a number of UK and Scottish government COVID-19 advisory bodies—SPI-M, MHRA Covid Vaccines Benefit and Risk Group, Scottish Government Covid Advisory Group and Chief Nursing Officer Nosocomial Advisory Group. DTB has served on a number of UK and Northern Ireland government COVID-19 advisory bodies. All other authors report no competing interests. RAL was a member of the Welsh Government COVID-19 Technical Advisory Group. RKO is a member of the National Institute for Health and Care Excellence (NICE) Technology Appraisal Committee, member of the NICE Decision Support Unit (DSU), and associate member of the NICE Technical Support Unit (TSU). RKO has served as a paid consultant to the pharmaceutical industry, providing unrelated methodological advice generally. She reports teaching fees from the Association of British Pharmaceutical Industry (ABPI) and the University of Bristol. SdeL has received grants through his University for vaccine related research from AstraZeneca, GSK, Moderna, MSD, Sanofi and Seqirus. He has also been a member of advisory boards for AstraZeneca, GSK, Sanofi and Seqirus, with any funding paid to his University. The remaining authors have nothing to declare.
