## [Peer Review File · Nature Communications]

Uptake of COVID-19 vaccinations amongst 3,433,483 children and young people: meta-analysis of UK prospective cohortsREVIEWER COMMENTS

Reviewer #1 (Remarks to the Author):

Thank you for the opportunity to review the work by Aldridge, et al. The authors use national data from the UK to describe early COVID-19 vaccine uptake among children and adolescents aged 5-17 years. While the population studied is novel, there are some limitations to the analysis. Please see my review below.

What are the noteworthy results?

- The authors investigate COVID-19 vaccine uptake among children and adolescents across the UK and examine specific factors associated with uptake in this large pediatric population. Strengths of the study include the robust national data sources, population size, and the novel look at this question in this population. The study's major limitations include the short interval of the study period overlapping the 5-11-year-old vaccine recommendation, and the restriction to factors available from all four UK data sources.

Will the work be of significance to the field and related fields? How does it compare to the established literature? If the work is not original, please provide relevant references.

- The topic covered is not new, but the work does cover a novel population – I was unable to identify existing studies of pediatric uptake in the UK. Most existing work in this arena is from the US, but there are a scattering of other countries represented. The authors included the paper by Khatatbeh, et al. but did not include relevant work from the US and Brazil; sample references provided below.

References (not exhaustive):

USA, pediatric/adol coverage

- <https://www.ncbi.nlm.nih.gov/pmc/articles/PMC8911999/>
- <https://www.cdc.gov/mmwr/volumes/71/wr/mm7126a3.htm>
- <https://pubmed.ncbi.nlm.nih.gov/34473680/>

USA, pediatric coverage disparities

- <https://www.ncbi.nlm.nih.gov/pmc/articles/PMC9045440/>
- <https://www.sciencedirect.com/science/article/pii/S0264410X22014037>

Brazil, determinants of uptake

- <https://www.sciencedirect.com/science/article/pii/S0033350622003079>

USA, vaccination status and reasons (survey)

- <https://www.tandfonline.com/doi/full/10.1080/07853890.2022.2045034>
- <https://www.sciencedirect.com/science/article/pii/S0033350622001597>
- <https://www.cdc.gov/mmwr/volumes/72/wr/mm7201a1.htm>

- Does the work support the conclusions and claims, or is additional evidence needed?

- Yes

Are there any flaws in the data analysis, interpretation and conclusions?

- While acknowledged by the authors, the staggered roll out schedule, specifically the late roll out for 5-11-year-olds relative to the study period, presents a major limitation and impacts the meaningfulness of the results. Most notably, the combined-age uptake data [e.g., Results/paragraph 1] is difficult to interpret due to the temporal issues at play.

Do these prohibit publication or require revision?

- Yes, given the short amount of time 5-11-year-olds were vaccine eligible within the study period, the lower uptake is expected. Roll out in England was 4/4/22, nearly eight months after the 16-17-year-old recommendation, and overlapping for less than 2 months with the study period. The authors have attempted to account for the time element with modeling, but limitations remain.

Is the methodology sound? Does the work meet the expected standards in your field?

- Yes

Is there enough detail provided in the methods for the work to be reproduced?

- Likely, but the individual data are not publicly available.

Minor comments

- Discussion/Page 6, 4th full paragraph: The date as listed as August 4, 2020 – should this be 2021?
- Figure 3 is not referenced in the results, only in the discussion. What is being displayed on the x-axis is also unclear – I understand that it's days, but why 300 days? Re: Figure 3a, the entire study period is longer than 300 days, and the 5–11-year-olds were not eligible for 300 days. [Yet, there appears to be dose 1 uptake starting at day 0 in that age group, and the line continues to day 300.]

Reviewer #2 (Remarks to the Author):

Summary

This is a meta-analysis of covid-19 vaccine uptake among 5-17-yr-olds in England, Northern Ireland, Wales and Scotland. The study period is from August 4 2021 to May 31 2022. The authors present 1) uptake of 1st, 2nd and booster dose, and 2) analysis of select demographic determinants of uptake (age, sex, vaccination status of adults in household, number of people in household). They conclude that uptake was low and diminished with subsequent doses, that younger children were less likely to get vaccinated than older, and that household vaccination status was associated with uptake.

I enjoyed reading the paper. It was easy to read, results were nicely presented and data carefully analysed. However, the results are not particularly novel or surprising.

Major comments

- 1) What differentiates the uptake results from those I am assuming have been available from national dashboards according to age and sex?
- 2) The inclusion of more demographic information such as region, socio-economic status, level of education, ethnicity etc would have improved the usability of the results. Why choose to do a meta-analysis and being restricted to a few common demographic covariates (lines 352-353) instead of a single country analysis with richer data? Statistical power is not really an issue here.
- 3) The use of a multi-state model to model the impact of covariates on the uptake taking competing events infection and death into account (line 262): It is appreciated, but hazard ratios are not so easy to communicate as e.g. uptake differences in percentage points. I agree that if you had near-complete ascertainment of infection it would be valuable to take competing risks into account. My understanding is that you do not have near-complete ascertainment (lines 270-273). Do you have an estimate of how many infections were recorded. To me this seems to be a very important determinant for uptake. Infection will not only postpone vaccination, but will often probably cancel.

Minor comments

- 4) Introduction: Could you elaborate on how vaccinations of CYP took place in the UK?
- 5) Results: I am assuming the recommendations did not include boosters for 5-11, why the uptake of the booster appears quite low in this age group.
- 6) Figure 3: What is the time scale. Time since eligibility? Time since August 4 2021?
- 7) Why was a meta-analysis needed instead of a combined analysis?
- 8) Why only include infections within 28-days of the study start on August 4 2021? (line 308)
- 9) Any supporting documentation that the 33% sample used for the English analyses are representative? (line 324)

- 10) Table 2: There is something wrong with the first country label.
- 11) Figure 3: How are the cumulative incidences derived? Do they take competing risks into account.
- 12) Consider providing sample R-code for the specification of the multi-state models.
- 13) Please label the transitions with names instead of numbers throughout.
- 14) Table 1: Please also provide a column for infections.

Reviewer #3 (Remarks to the Author):

The report presents data on 3,433,483 UK children. It presents and discusses vaccination uptake for first dose, second dose and booster, also with a subset for each UK nation. Results are also contextualised with respect to:

- Roll-out age ranges
- Sex
- Vaccination status in household
- Household size

The manuscript is very well organised and written. However, I have remarks:

MAJOR:

- Introduction. This section in my opinion misses an important contextual point: the difference in overall mortality and morbidity of the different SARS-CoV-2 variants up to 16th February (and even further up to 31st May) 2022, which intuitively have been a moderating factor in the decision of whether to vaccinate or not own children.
- Introduction. Also, the difference in overall mortality and morbidity of SARS-CoV-2 at different paediatric ages is not appropriately described, although it was very much emphasised by the media.
- Beyond the introduction, the two factors mentioned above are not considered in the data analysis either. Moreover, incidence of children who got infected before potential vaccination was evaluated in terms of governmental recommendation not to vaccinate within 28 days, but not in terms of acquired immunity. This is not irrelevant, as overall the discussion emphasizes the association between (household) vaccine hesitancy and young ages. The statement at line 217 "These findings are in line with research investigating intentions regarding COVID-19 vaccination of CYP from both CYP themselves and parents in the UK and globally, which have also shown that vaccine hesitancy increased inversely with age (15-17)." might miss the influence of time-dependent variations in SARS-CoV-2 morbidity and mortality, age-dependent differences in morbidity and mortality not evaluated here (but reported by the media and likely influencing decisions), and cumulative data on children's immunity (i.e., number of already infected at the time of vaccination offer, for whom families might have concluded vaccination was not needed within short).

This data is undoubtedly very interesting to the governmental bodies and UK institutions. However, a concrete risk of overemphasizing families' vaccine hesitancy is embedded in data analysis and presentation at present. In my opinion, a reanalysis considering time-variant / time-dependent factors is needed, to minimise the risk of injecting partial interpretations in scientific literature. Also, an attempt to generalise conclusions should be made, for example by comparing the data with a second database from a country which followed a different roll-out schedule (eg. Israel).

MINOR:

- Introduction, lines 73-76 "Following the initial success of the COVID-19 vaccine programme in adults and evidence from trials and observational studies on the benefits and risks of vaccines in CYP (11), the target population broadened to include younger adults with no underlying health conditions who were not living with vulnerable people, before including all CYP." The sentence is equivocal in my opinion, as it sounds that living with vulnerable people was an exclusion factor for vaccination, which was clearly not the case. Please amend.

Thank you for the invitation to further revise our paper. We are very grateful to the Reviewers for their additional reviews, which we have as a team very carefully considered. Our responses to the points raised are detailed below. For your convenience, we first reproduce the feedback received verbatim before detailing our responses.

Reviewer #1

Comment 1: *The topic covered is not new, but the work does cover a novel population – I was unable to identify existing studies of pediatric uptake in the UK. Most existing work in this arena is from the US, but there are a scattering of other countries represented. The authors included the paper by Khatatbeh, et al. but did not include relevant work from the US and Brazil; sample references provided below.*

References (not exhaustive):

- USA, pediatric/adol coverage
 - <https://www.ncbi.nlm.nih.gov/pmc/articles/PMC8911999/>
 - <https://www.cdc.gov/mmwr/volumes/71/wr/mm7126a3.htm>
 - <https://pubmed.ncbi.nlm.nih.gov/34473680/>
- USA, pediatric coverage disparities
 - <https://www.ncbi.nlm.nih.gov/pmc/articles/PMC9045440/>
 - <https://www.sciencedirect.com/science/article/pii/S0264410X22014037>
- Brazil, determinants of uptake
 - <https://www.sciencedirect.com/science/article/pii/S0033350622003079>
- USA, vaccination status and reasons (survey)
 - <https://www.tandfonline.com/doi/full/10.1080/07853890.2022.2045034>
 - <https://www.sciencedirect.com/science/article/pii/S0033350622001597>
 - <https://www.cdc.gov/mmwr/volumes/72/wr/mm7201a1.htm>

Response: Thank you for highlighting these additional references, which we have now incorporated into the revised Introduction. The revised text (paragraph 2) now reads.

"The gradual rollout of the COVID-19 vaccine to progressively younger age groups of adults coincided with a steady decrease in uptake. Originally targeting the elderly and those with underlying health conditions, uptake was as high as 96% (80 years old and above receiving their primary schedule) in England, but declined to 70% in 18-19-year-olds (14). Vaccination in CYP is complex and tightly bound to issues regarding consent, autonomy and access (14). Within the UK, COVID-19 vaccine uptake in CYP has been shown to vary in state-school pupils with region and ethnicity with a slower uptake in pupils who spoke English as a second language, attended special needs schools, received free school meals or lived in more deprived areas (15). Outside of the UK, studies have shown that uptake of the COVID-19 vaccine in CYP can be influenced by several factors, including availability (16), region (17,18), age (17–19), ethnicity (17,18,20), immigration status (19), urbanicity (21), vulnerability (22), education (18), household income (18,19), parental vaccination status (18), prior COVID-19 diagnosis (18) and parents age (18)."

Comment 2: *While acknowledged by the authors, the staggered roll out schedule, specifically the late roll out for 5-11-year-olds relative to the study period, presents a major limitation and impacts the meaningfulness of the results. Most notably, the combined-age uptake data [e.g., Results/paragraph 1] is difficult to interpret due to the temporal issues at play. Yes, given the short amount of time 5-11-year-olds were vaccine eligible within the study period, the lower uptake is expected. Roll out in England was 4/4/22, nearly eight months after the 16–17-year-old recommendation, and overlapping for less than 2 months with the study period. The authors have attempted to account for the time element with modelling, but limitations remain.*

Response: Thank you for this comment. We agree that the staggered roll out schedule makes the data challenging to analyse and interpret. We considered a priori that a longer study period to capture more of the vaccine uptake for 5-11-year-olds would improve interpretation for this group, however data was only available from all four nation until May 31st. We have nonetheless now performed a sensitivity analysis investigating the effect of extending our study period to the end of December 2022. The results of the nations whose governance allowed this extension (i.e., Northern Ireland, Scotland and Wales) are now reported in the Supplementary material (Please see Tables 4-7 and Figures 4 and 5) in the form of a sensitivity analysis. We found very little change in the vast majority of the coefficients produced from the meta-analysis with all point estimates of the original

study falling within the 95% confidence intervals (CIs) of the updated analysis. For example, for 5-11-year-olds the aHR (95%CI) for first and second vaccine was 0.10 (0.06-0.19), and 0.54 (0.37-0.80) for the original analysis with an end date in May 2022. The updated analysis with an end date of December 2022 resulted in aHRs of 0.05 (0.02-0.14), 0.5 (0.42-0.59) and 0.01 (0-0.2) for the 1st, 2nd and booster doses, respectively.

Minor Comments:

1. *Discussion/Page 6, 4th full paragraph: The date as listed as August 4, 2020 – should this be 2021? – Response: Yes, thank you - this has now been corrected.*
2. *Figure 3 is not referenced in the results, only in the discussion. What is being displayed on the x-axis is also unclear – I understand that it's days, but why 300 days? Re: Figure 3a, the entire study period is longer than 300 days, and the 5–11-year-olds were not eligible for 300 days. [Yet, there appears to be dose 1 uptake starting at day 0 in that age group, and the line continues to day 300.]*

Response: The original figure was to represent the study period between 4th August 2021 and 31st May 2022, which is 300 days exactly. This figure has been updated (along with the similar figure in the supplementary material) to show dates rather than number of days. The scale runs from August 2021 to June 2022, and demonstrates uptake throughout the whole study period. While the younger age groups are ineligible, their uptake remains at 0. Figure 3 is now also referenced in the Results.

Reviewer #2

Comment 1: *What differentiates the uptake results from those I am assuming have been available from national dashboards according to age and sex?*

Response: Our results are different from those available through national dashboards as we have been able to take into account ineligibility for vaccination in CYP following infection at the individual level as a competing risk in the model. National dashboards typically report population level infection rates, but due to a lack of linked data at the individual level, are unable to adjust analyses for infection as a competing risk for vaccine uptake. Moreover, our study has investigated a range of factors such as household size and the vaccination status of adults in the same household as a predictor of vaccination/non-vaccination. The Discussion emphasises this in the following statement:

"It is also the first application of a multi-state model to be used on COVID-19 vaccination uptake, allowing us to account for competing events such as COVID-19 infection, which delays vaccination by 28 days, and death."

The Introduction of the manuscript has also been revised to further emphasise these unique contributions.

"We aimed to investigate the uptake of the COVID-19 vaccines across all four UK nations following JCVI recommendation for CYP using prospective population-based cohort analyses on routinely collected electronic health record (EHR) data adjusting for infection as a competing risk, and to explore the association of vaccine uptake with age, sex, and household factors."

Comment 2: *The inclusion of more demographic information such as region, socio-economic status, level of education, ethnicity etc would have improved the usability of the results. Why choose to do a meta-analysis and being restricted to a few common demographic covariates (lines 352-353) instead of a single country analysis with richer data? Statistical power is not really an issue here.*

Response: Thank you for this comment. We wanted to include demographic information, however not all information was available in all countries owing to differing governance permissions. We were able to obtain deprivation and urbanicity data for Northern Ireland, Scotland and Wales, and ethnicity for Scotland and Wales. Deprivation, urbanicity and ethnicity were available in England, however, including these added variables in the model resulted in computational difficulties on the systems available owing to the scale of the data. We performed an analysis including these covariates for the available regions and have now included these results in our Supplementary material in the form of a sensitivity analysis (Please see Tables 4-7 and Figures 4 and 5). The results reported in the main text were robust when including these additional variables for the available regions. In all instances, the original aHR values fell within the 95%CI of the expanded analysis.

We chose to perform a meta-analysis across all 4 UK nations to provide a UK-wide perspective on uptake to identify heterogeneity between nations (of which we find evidence) and therefore provide insights for policy makers and national public health agencies.

Comment 3: *The use of a multi-state model to model the impact of covariates on the uptake taking competing events infection and death into account (line 262): It is appreciated, but hazard ratios are not so easy to communicate as e.g. uptake differences in percentage points. I agree that if you had near-complete ascertainment of infection it would be valuable to take competing risks into account. My understanding is that you do not have near-complete ascertainment (lines 270-273). Do you have an estimate of how many infections were recorded. To me this seems to be a very important determinant for uptake. Infection will not only postpone vaccination, but will often probably cancel.*

Response: We agree that it would be better to have complete and reliable infection data for this study at the individual level. LFT and RT-PCR results for SARS-COV-2 infection were the best indicators of infection at the individual level. We acknowledge that the reporting rate of LFTs is unknown and this is a potential limitation of the study. However, it remains important to account for competing risks when the competing risk is known, even in the presence of potentially missing data. We have added this point to the Discussion:

"The extent to which RT-PCR tests were used is unknown, particularly for CYP whose uptake of these tests may have been affected by the practicality of using them. The rate of reporting for positive lateral flow devices was also unknown. However, it remains important to account for competing risks when known, even in the presence of potentially missing data."

Furthermore, we have now also added the infection counts for each nation into Supplementary Table 1, and cumulative probability plots (Supplementary Fig 3) for each nation detailing the probability of infection, vaccination, or death throughout the study period.

Minor comments

1. *Introduction: Could you elaborate on how vaccinations of CYP took place in the UK?* The vaccination of Response: CYP used a combination of GP surgeries and vaccination centres. We have added the following sentence to the Introduction to make this clearer:
"On 4th August 2021, JCVI recommended including 16-17-year-olds who were not in a clinical risk group in the Pfizer-BNT162b2 vaccine schedule (12), followed by similar recommendations for 12-15-year-olds on 13th September 2021 (13) and 5-11-year-olds on 16th February 2022 (9). Each nation took time to review this advice before officially moving forward with the vaccination programme of each age group, making vaccinations available for these groups in General Practice surgeries and vaccination centres. In all cases, the schedule started within 0-2 days following JCVI recommendation for 16-17-year-olds, 1-34 days for 12-15-year-olds and 14-47 days for 5-11-year-olds (see Supplementary Fig 1 for nation-specific details)."
2. *Results: I am assuming the recommendations did not include boosters for 5-11, why the uptake of the booster appears quite low in this age group.*
Response: Our study period did not extend to the vaccination period for boosters in 5-11 year olds. We have now clarified this in the second results paragraph: *"5-11-year-olds had the lowest vaccine uptake, where 11% (192,994) received their first vaccine, and 0.2% (4,152) received their second vaccine. 5-11-year-olds did not become eligible for their booster dose before the conclusion of the study window".*
3. *Figure 3: What is the time scale. Time since eligibility? Time since August 4 2021?*
Response: The timescale is from August 4th 2021; we have updated Figure 3 to show dates to make this more clear.
4. *Why was a meta-analysis needed instead of a combined analysis?*
Response: A combined analysis would have been the ideal analysis to undertake; however, given the sovereign nature of the datasets used, it was not possible to share large, linked, individual level electronic health records across national jurisdictions. The data for each region were held and analysed in separate Trusted Research Environments, which could not be combined. In keeping with related previous analyses (e.g. Kerr et al, IJE <https://doi.org/10.1093/ije/dyac199>; Agrawal et al, Lancet [https://doi.org/10.1016/S0140-6736\(22\)01656-7](https://doi.org/10.1016/S0140-6736(22)01656-7)), we therefore chose to perform a meta-analysis, pooling the data across regions using the extracted coefficients. We have added a sentence to the Methods – Statistical analysis to make this clear:
"The derived aggregate data from each nation (reported as adjusted log hazard ratios (aHR) and corresponding standard errors) were exported from the respective TRES and meta-analysed in a two-stage individual participant data random effects meta-analysis (45). A pooled analysis could not be performed with the data available given the sovereign nature of the datasets used. It was not possible to share large, linked, individual level electronic health records across jurisdictions. In keeping with related, previous analyses (e.g. (46,47)) we therefore chose to perform a meta-analysis, pooling the data across regions using the extracted coefficients. Effect estimates were presented as hazard ratios and corresponding 95% confidence intervals. Restricted maximum likelihood estimation was used to estimate the pooled coefficients. Heterogeneity between the estimated effects between the nations were quantified using the between-study standard deviation. I-squared was used to assess the proportion of total variability owing to between-study heterogeneity."

5. *Why only include infections within 28-days of the study start on august 4 2021? (line 308)*
Response: This was to capture individuals entering the study in an infection/ post-infection state where they were ineligible to receive the vaccine. We have added the following sentence to the statistical analyses to clarify this point *"This ensured that individuals who were not eligible to receive vaccination owing to recent SARS-CoV-2 infection (i.e., infection was a competing risk for vaccine upon study entry) were adjusted for in the model."*
6. *Any supporting documentation that the 33% sample used for the English analyses are representative? (line 324)*
Response: We have added this reference to support this statement in the methodology: <https://bmjopen.bmj.com/content/6/4/e011092.short>
7. *Table 2: There is something wrong with the first country label.*
Response: Thank you, this has been corrected.
8. *Figure 3: How are the cumulative incidences derived? Do they take competing risks into account.*
Response: Yes, the cumulative incidences were derived from the multi-state model accounting for competing risks. We have added the following sentence to the Statistical Analysis section to clarify: *"Cumulative incidence of first, second and booster dose were calculated from the multistate models, having adjusted for infection and mortality as competing risks."*
9. *Consider providing sample R-code for the specification of the multi-state models.*
Response: The code used for the analysis will be made freely available on the project GitHub page (<https://github.com/HDRUK/DaCVaP>) when the manuscript is published.
10. *Please label the transitions with names instead of numbers throughout.*
Response: We have made this change throughout.
11. *Table 1: Please also provide a column for infections.*
Response: Since infection is time-varying, for example, individuals could move in and out of the infection state in the multistate model at repeated times over the follow-up, it is less informative to present infection as a static health state in this table. Rather we have presented the health state occupancy of the infection state over time in the supplementary Figure 3, and the total number of people who became infected during the study period for each nation in Supplementary table 1.

Reviewer #3

Comment 1: *Introduction. This section in my opinion misses an important contextual point: the difference in overall mortality and morbidity of the different SARS-CoV-2 variants up to 16th February (and even further up to 31st May) 2022, which intuitively have been a moderating factor in the decision of whether to vaccinate or not own children.*

Response: We have added a paragraph to the Introduction to introduce this idea and further elaborate in the Discussion to discuss this potential issue, which we describe in more detail in response to comment 3 below. The Introduction now includes:

"It is known that overall mortality and morbidity differed with alternative dominant variants (24). In the UK, the dominant variant from 1st August 2021 to 19th December 2021 was the Delta variant and from 20th December 2021 onwards was the Omicron variant (25)."

And the following paragraph has been added to the Discussion:

"An additional potential factor that may have influenced vaccine uptake was the underlying COVID-19 morbidity and mortality in CYP and/or the general population. It is known that COVID-19 morbidity and mortality varied with different dominant variants (24). During this study period in the UK, the Delta and Omicron variants were dominant. However, the Delta variant was only dominant for a maximum of 4.5 months from the beginning of the vaccine roll out programme. For this reason, it was not possible to investigate the impact of dominant variant on vaccine uptake owing to low event rates. For 5-11 year olds for example, the vaccine roll out period was only applicable during the Omicron dominant period."

Comment 2: *Introduction. Also, the difference in overall mortality and morbidity of SARS-CoV-2 at different paediatric ages is not appropriately described, although it was very much emphasised by the media.*

Response: We have added a paragraph to the Introduction to describe the mortality rate at different paediatric ages. The introduction now reads:

"In the US, it has been shown that during the time period of 1st August 2021 to 31st July 2022, COVID-19 was a leading cause of death in CYP. Crude death rates were estimated as 0.4 per 100,000 CYP aged 5 to 9 years, 0.5 per 100,000 CYP aged 10 to 14 years, and 1.8 per 100,000 for CYP aged 15 to 19 years (23)."

Comment 3: *Beyond the introduction, the two factors mentioned above are not considered in the data analysis either. Moreover, incidence of children who got infected before potential vaccination was evaluated in terms of governmental recommendation not to vaccinate within 28 days, but not in terms of acquired immunity. This is not irrelevant, as overall the discussion emphasizes the association between (household) vaccine hesitancy and young ages. The statement at line 217 "These findings are in line with research investigating intentions regarding COVID-19 vaccination of CYP from both CYP themselves and parents in the UK and globally, which have also shown that vaccine hesitancy increased inversely with age (15–17)." might miss the influence of time-dependent variations in SARS-CoV-2 morbidity and mortality, age-dependent differences in morbidity and mortality not evaluated here (but reported by the media and likely influencing decisions), and cumulative data on children's immunity (i.e., number of already infected at the time of vaccination offer, for whom families might have concluded vaccination was not needed within short).*

Response: We have now made clearer that morbidity and mortality were explicitly modelled as health states in the model in the form of infection and mortality states. This will account for variations in morbidity and mortality in CYP (and account for age-dependent difference in morbidity and mortality since age is fitted as a covariate in the model). The number already infected is captured in the model by including infection as a potential entry state for CYP who are infected prior to cohort start. It would not be possible to adjust the data analysis for dominant variants, either by including variant as a covariate in the model or stratifying the analyses, since the dominant variant in the UK between 4th August 2021 (when the first JCVI approval in CYP was granted) and 19th December 2021 was Delta, and Omicron thereafter. Due to low numbers of events in the maximum of 4.5 months of the dominant Delta variant (and 5–11-year-olds who were only eligible during the Omicron dominant period) there was insufficient power to detect any potential difference. We have added to the Discussion to stress this point, which now reads:

"An additional potential factor that may have influenced vaccine uptake was the underlying COVID-19 morbidity and mortality in CYP and/or the general population. It is known that COVID-19 morbidity and mortality varied with different dominant variants (24). During this study period in the UK, the Delta and Omicron variants were dominant. However, the Delta variant was only dominant for a maximum of 4.5 months from the beginning of the vaccine roll out programme. For this reason, it was not possible to investigate the impact of dominant variant on vaccine uptake owing to low event rates. For 5-11 year olds for example, the vaccine roll out period was only applicable during the omicron dominant period."

This data is undoubtedly very interesting to the governmental bodies and UK institutions. However, a concrete risk of overemphasizing families' vaccine hesitancy is embedded in data analysis and presentation at present. In my opinion, a reanalysis considering time-variant / time-dependent factors is needed, to minimise the risk of injecting partial interpretations in scientific literature. Also, an attempt to generalise conclusions should be made, for example by comparing the data with a second database from a country which followed a different roll-out schedule (eg. Israel).

Response:

We agree that ideally, we would fit vaccination status of the household as a time-varying covariate in the model. However, this approach is computationally impractical given the scale, and nature of the data. As noted in response to Reviewer 2 Comment 2, adding additional covariates in the model led to computational issues when applied to the dataset in England, particularly with respect to available memory. We have added the following to the Discussion to address this point:

"A further limitation of the study is the inability to account for household vaccination status as a time-varying covariate; owing to the scale and nature of the data, it would be computationally impractical to run such analyses across all four nations."

We agree that an external validation in other countries with an alternative roll-out schedule would be helpful. Unfortunately, accessing individual level data in these countries is difficult, particularly with respect to governance constraints across borders. We have added to the Discussion to emphasise that further work is required in other settings with alternative roll-out strategies.

"This is the first study to use pooled data from all four UK nations to assess vaccine uptake in CYP. All nations used common definitions and methods, which improved the consistency of the pooled meta-analysis results. It is also the first application of a multi-state model to be used on COVID-19 vaccination uptake, allowing us to account for competing events such as COVID-19 infection, which delays vaccination by 28 days, and death. By assigning a universal start date to all nations defined

by the JCVI recommendation, we have excluded many individuals who would have received their vaccines early based on the assumption that these individuals were more likely to have underlying health conditions and/or reside in clinically vulnerable households. Future work could explore vaccine uptake in CYP with alternative vaccine roll-out strategies.

Minor comments:

1. *Introduction, lines 73-76 "Following the initial success of the COVID-19 vaccine programme in adults and evidence from trials and observational studies on the benefits and risks of vaccines in CYP (11), the target population broadened to include younger adults with no underlying health conditions who were not living with vulnerable people, before including all CYP." The sentence is equivocal in my opinion, as it sounds that living with vulnerable people was an exclusion factor for vaccination, which was clearly not the case. Please amend.*

Response: Thank you for pointing this out; this sentence has now been amended to: "Following the initial success of the COVID-19 vaccine programme in adults and evidence from trials and observational studies on the benefits and risks of vaccines in CYP (11), the target population broadened to include younger adults with underlying health conditions and those who were living with vulnerable people, before finally including all CYP."

The opportunity to revise our paper in the light of this constructive feedback has helped us to improve the quality of our work for which we are grateful. We trust that these revisions are to your satisfaction and that our paper is now considered suitable for publication. Please don't however hesitate to contact us if further clarification or revisions are required.

With kind regards,

Dr Sarah J Aldridge

REVIEWERS' COMMENTS

Reviewer #2 (Remarks to the Author):

Thank you for the revision which has been accomodating to my comments. Supp figure 3 is a nice addition (in the legend, mode should be model).

Nothing else to add.

Reviewer #2 (Remarks on code availability):

The authors state that code will be available upon publication of the manuscript.

Reviewer #3 (Remarks to the Author):

My comments have been addressed.

Reviewer #3 (Remarks on code availability):

The code is frankly very complicated to access, even before checking the models. It is a collection of scripts, but - for example - scripts for Scotland do not correspond to those for Wales. The good looks more in the development phase, rather than a usable library.